# Monotonic and Fatigue Properties of Steel Material Manufactured by Wire Arc Additive Manufacturing

**Michael Wächter [1],\*** , **Marcel Leicher [2]**, **Moritz Hupka [1]**, **Chris Leistner [3]**, **Lukas Masendorf [1]**, **Kai Treutler [2]** , **Swenja Kamper [2]**, **Alfons Esderts [1]**, **Volker Wesling [2]** and **Stefan Hartmann [3]**

[1] Institute for Plant Engineering and Fatigue Analysis, Clausthal University of Technology, Leibnizstraße 32, 38678 Clausthal-Zellerfeld, Germany; moritz.hupka@tu-clausthal.de (M.H.); lukas.masendorf@tu-clausthal.de (L.M.); alfons.esderts@tu-clausthal.de (A.E.)

[2] Institute of Welding and Machining, Clausthal University of Technology, Agricolastraße 2, 38678 Clausthal-Zellerfeld, Germany; leicher@isaf.tu-clausthal.de (M.L.); treutler@isaf.tu-clausthal.de (K.T.); swenja.kamper@tu-clausthal.de (S.K.); volker.wesling@tu-clausthal.de (V.W.)

[3] Institute of Applied Mechanics, Clausthal University of Technology, Adolph-Roemer-Straße 2A, 38678 Clausthal-Zellerfeld, Germany; chris.leistner@tu-clausthal.de (C.L.); stefan.hartmann@tu-clausthal.de (S.H.)

\* Correspondence: michael.waechter@tu-clausthal.de

**Abstract:** In this study, the monotonic and cyclic material properties of steel material of medium static strength produced additively in the wire arc additive manufacturing (WAAM) process were investigated. This investigated material is expected to be particularly applicable to the field of mechanical engineering, for which practical applications of the WAAM process are still pending and for which hardly any characteristic values can be found in the literature so far. The focus of the investigation was, on the one hand, to determine how the material characteristics are influenced by the load direction in relation to the layered structure and, on the other hand, how they are affected by different interlayer temperatures. For this purpose, monotonic tensile tests were carried out at room temperature as well as at elevated temperatures, and the cyclic material properties were determined. In addition, the hardness of the material and the residual stresses induced during production were measured and compared. In addition to the provision of characteristic properties for the investigated material, it was aimed to determine the extent to which the interlayer temperature influences the strength characteristics, since this can have a considerable influence on the production times and, thus, the economic efficiency of the process.

**Keywords:** wire arc additive manufacturing; medium strength steel; tensile properties; cyclic properties; strain-controlled tests; interlayer temperature

## 1. Introduction

Efforts to manufacture components in a more resource-efficient manner, and thus, ultimately more economically, mean that additive manufacturing processes are increasingly becoming the focus of application, even for safety-relevant components. Reviews on the additive manufacturing of metals and the classification of different additive manufacturing processes can be found, for example, in [1,2]. One of these additive manufacturing processes, which is expected to produce similarly high component strengths as conventionally manufactured components, is wire arc additive manufacturing (WAAM; see [3]). For special applications, such as turbine blades made of titanium or high-grade steels, production with WAAM is already established and accepted, see e.g., [4]. Applications can also be found in the aerospace industry, [2], in marine and architecture, as well as in nuclear power

technology, [1]. For mechanical engineering applications, however, large-scale use is still pending. This can be explained by the fact that the manufacturing process is still comparatively expensive due to long production times, and accordingly, high unit costs. For this reason, the application becomes profitable when producing small quantities or prototypes or when using high-priced materials as well as materials that are difficult to handle by machining or casting.

Another trend that has become apparent is the greater variety of machines, systems, and vehicles that are precisely tailored to the wishes of the customer. If this trend is more consistently thought through to the end, in certain cases, a necessity arises to switch from series to single-part production, even at the component level. This is also known as mass customization, [5]. In order to make the WAAM process usable for this broad application as well, the following two aspects have to be considered:

1.  The process needs to be applicable with the classic materials of mechanical engineering, namely steel, and for manufacturing safety-relevant components. For this to be the case, the essential material properties must be known.
2.  To be economically applicable, the process must be optimized in such a way that short cycle times, and therefore, low unit costs result. At the same time, however, the strengths that can, in principle, be achieved must not be reduced significantly.

In addition to the central properties of monotonic strength, the fatigue strength properties of safety-relevant components have to be verified. This has only been documented to a very limited extent or for individual cases of WAAM materials and components in the literature so far, see, e.g., [6,7]. With this study, an attempt shall be made to compare the characteristic values found for one exemplary WAAM material with those of conventional rolled, forged, or cast steel materials and thereby contribute to the understanding of WAAM material.

Largely unknown is the influence of the interlayer temperature on the WAAM material in question. The interlayer temperature is the temperature to which the last layer produced cools down before the next layer is applied. This definition corresponds to the use of Li et al. [8]. Another quantity to describe the cooling conditions in a layer is the interlayer cooling time, as used, for example, by Lee et al. [9].

On the one hand, a high interlayer temperature leads to slow cooling of the material, since the temperature of the entire component is kept at a high level throughout the whole production process. Based on the experience from conventional weld seams, this suggests that monotonic strengths (tensile strength $R_m$ and yield strength $R_p$) tend to be lower than at a low interlayer temperature that is associated with a high cooling rate. On the other hand, it can be expected that a high interlayer temperature will lead to a better bond between the different layers as well as to low distortion or residual stresses.

In this paper, the microstructure as well as the monotonic tensile loading and fatigue properties of a WAAM material are investigated for the following examples:

1.  Produced from one exemplary welding filler material normally used for welding base materials of tensile strengths of about 650 MPa,
2.  For which the interlayer temperature was varied during different manufacturing lots.

The influence of the interlayer temperature on the strength of the material as well as on the residual stresses within the manufactured material shall be investigated. Hereby, the first statements about the possibility of reducing the manufacturing time while maintaining or optimizing the component properties are to be derived.

## 2. Specimen Materials and Experiments

In this section, firstly, the manufacturing of the WAAM material under consideration is described. Based on this, secondly, the used specimens that were extracted from the aforementioned raw material are presented and, thirdly, the conducted experiments are described.

### 2.1. Manufacturing and Preparation of Specimen Material

To investigate the above-mentioned influences, cuboids were manufactured by the WAAM process, Figure 1a. These consisted of 40 layers, each approximately 2.8 mm thick. A metal inert gas (MIG) welding robot was used for production (see Table 1 for process parameters). The welding filler material was ISO 14341-A-G 50 7 M21 4Mo in accordance with ISO Standard 14341 [10]. In conventional welding, it is used to generate weld material with a minimum yield strength of $R_{p,min}$ = 500 MPa, a tensile strength in the range of $R_m$ = 560–720 MPa, and minimum elongation at rupture of $A_{min}$ = 18%. The filler material is usually used to weld low-alloyed high strength structural steels with a correspondent strength of about 560 MPa.

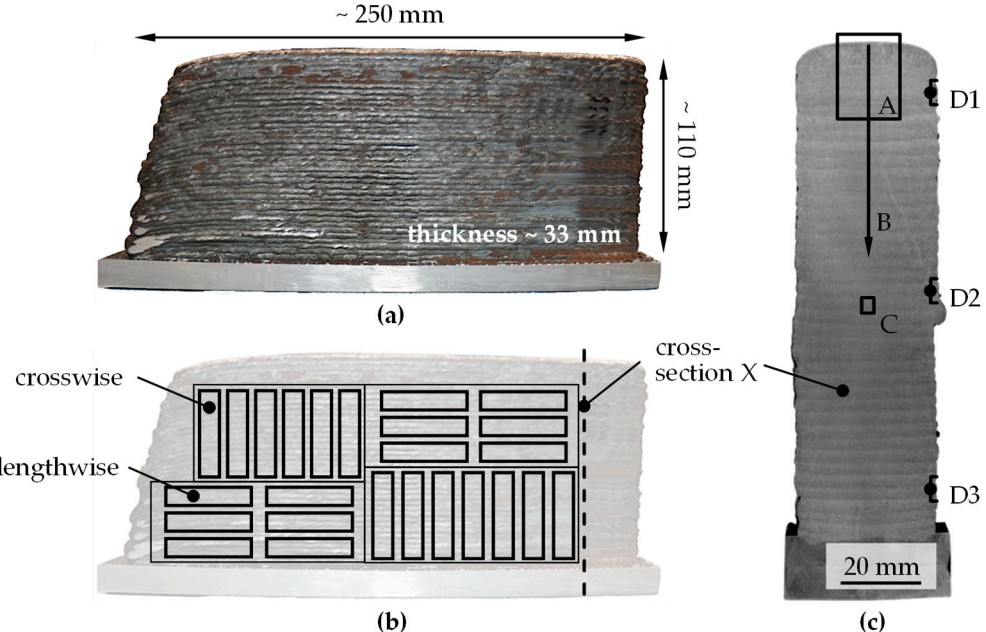

**Figure 1.** Wire arc additive manufacturing (WAAM) material: (**a**) semi-finished block products, (**b**) sampling plan for tensile specimens with directional designation related to the weld layers, (**c**) regions for micrographs (A and C), path for hardness measurements (B), and points for residual stress determinations (D1 to D3).

**Table 1.** Parameters of the manufacturing process.

| Welding Parameters. | Value |
|---|---|
| Welding Current | 112 A |
| Voltage | 23.1 V |
| Wire Feed | 4.5 $\frac{m}{min}$ |
| Wire-⌀ | 1.2 mm |
| Inert Gas Quantity | 12 $\frac{l}{min}$ |
| Width of Pendulum Movement | 24 mm |
| Frequency of Pendulum Movement | 0.7 Hz |
| Feed | 50 $\frac{mm}{min}$ |
| Path Velocity | 2.1 $\frac{m}{min}$ |

Welding was carried out with a pendulum movement in order to produce the largest possible weld pool and to enable adequate melting of the deposited layer. Thus, a good connection of the newly formed layer to the previous one should be ensured. A large energy input was used to achieve high melting rates and thus shorter manufacturing times (process optimization mentioned previously).

In this way, WAAM material was produced using two different interlayer temperatures, 150 and 300 °C. It was measured at three points distributed over the layer with sheath tip on thermocouples in

combination with a hand-held device for data acquisition. Thereby, temperature differences between the beginning and end of a layer of maximum 20 °C resulted. A recording of the temperature at the three thermocouples throughout the entire manufacturing process was not made, as the thermocouples had to be reattached after each layer applied to avoid their destruction. Due to the shorter waiting times between layer application for the interlayer temperature of 300 °C, the production of these cuboids took about a factor of 2 shorter compared to the interlayer temperature of 150 °C.

To prepare the manufactured material for further examination, first, measurements of residual stresses were taken on some of the semi-finished product cuboids. Three measuring points per cuboid were distributed over the cuboid's height (see D1 to D3 in Figure 1c). After that, the semi-finished product cuboids were sawn to obtain cross-section X in Figure 1b,c in order to carry out microstructural investigations (see regions A and C in Figure 1c) and hardness measurements along path B in Figure 1c.

The cuboids were divided into quadrants, Figure 1b, from which sample blanks were extracted lengthwise and crosswise to the welding direction. From these, the following unnotched tensile specimens were taken by CNC (Computerized Numerical Control) machining:

1. Fatigue specimens with a diameter in the test range of 3 mm. The surfaces of the specimens in the test area were polished, Figure 2a;
2. Proportional tensile specimens with a diameter in the test range of 5 mm for monotonic tensile tests at room temperature, according to [11], Figure 2b;
3. Tensile specimens with a diameter in the test range of 8 mm for monotonic tensile tests at elevated temperatures according to [12], Figure 2c. In addition,
4. flat tensile specimens with a cross-section of 10 by 2.1 mm for measuring the local deformation behavior with digital image correlation, Figure 2d, were extracted from the raw cuboids. Technical drafts with complete measures can be found throughout the Supplementary Materials at the end of this paper.

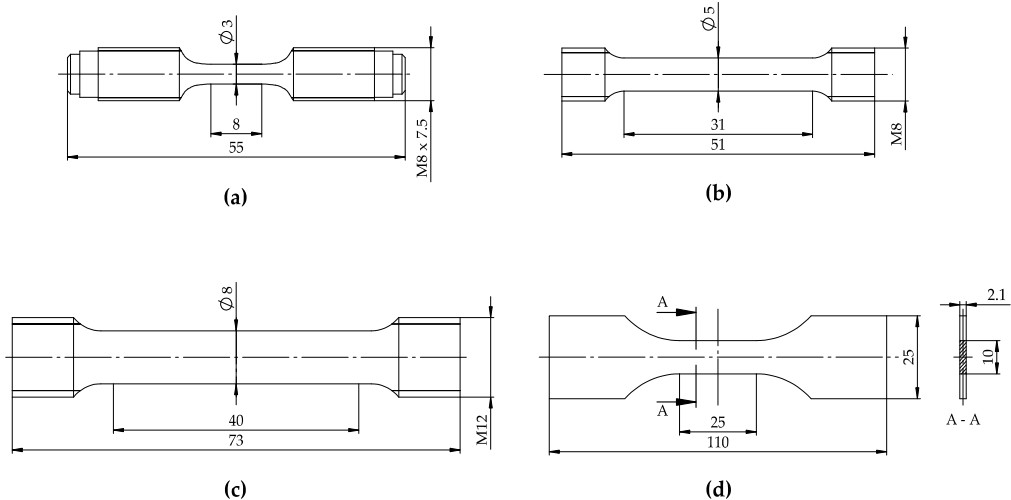

**Figure 2.** Technical drafts of tensile specimens with essential dimensions: (**a**) fatigue specimen, (**b**) specimen for tensile tests at room temperature, (**c**) specimen for tensile tests at elevated temperatures, (**d**) specimen for deformation measurements with digital image correlation.

## 2.2. Conducted Measurements and Experiments

In the following text, the procedures used for the different tests and measurements are listed and described briefly.

To gain insight into the residual stress state in the WAAM material, measurements were made using the hole-drilling strain gauge method. To apply the necessary strain gauge rosettes, 2 to 3 mm of the rough weld surface had to be removed by milling. The strain gauge rosettes used were of type

HBM 1-RY61-1.5/120K and the hole diameter was 1.8 mm. The strain was measured over a hole depth of 0.72 mm, and the residual stress was evaluated using the procedure described in ASTM Standard E837 [13].

In accordance with Vickers, the micro hardness HV 0.2 was measured every 0.3 mm along path B in agreement with ISO Standard 6507-1 [14]. The small test load for the Vickers hardness test was chosen to acquire a detailed hardness profile and to be able to determine different zones within the welded material. In addition, micrographs were prepared by polishing and nital etching the extracted material.

While the monotonic properties at room temperature were determined by tensile tests according to ISO Standard 6892-1 [15], the tests at elevated temperatures, namely at 150 and 300 °C, were conducted in accordance with ISO 6892-2 [12]. Both types of tests are conducted on testing machines for tensile testing of type Z100 in combination with control software testXpert, both from Zwick GmbH & Co. KG (Ulm, Germany). The testing machine used for the tests at elevated temperatures is equipped with a heating chamber, also from Zwick. First, all tensile specimens to be tested were placed in an external oven (power: 5 kW, maximum temperature: 850 °C) for a sufficient dwell time to heat the specimens to the core. Successively, the samples were removed from the oven and clamped in the testing machine surrounded by the pre-heated chamber. After the chamber had been closed around the specimen, the test started. While the tests at room temperature could be conducted using a clip-on extensometer, such a sensor was not available for the testing machine for elevated temperatures. Hence, the crosshead movement was used to calculate the necessary strain for the evaluation of the yield strength $R_{p0.2}$ and elongation at rupture A. Due to the inaccuracies associated with the use of the crosshead movement the elongation at rupture was crosschecked with measurements on the broken specimens.

While fatigue testing and tensile testing at room temperature was conducted on specimens from both extraction directions, tensile properties at elevated temperature were only determined for the crosswise extraction direction due to limited available material.

To describe the cyclic fatigue and deformation behavior, the cyclic properties were determined. The cyclic properties $\sigma'_f$, b, $\varepsilon'_f$, and c together with the elastic modulus E are used to describe the strain–life curve in accordance with Coffin and Manson, [16,17], Equation (1):

$$\varepsilon_a = \varepsilon_{a,el} + \varepsilon_{a,pl} = \frac{\sigma'_f}{E}(2N)^b + \varepsilon'_f(2N)^c \tag{1}$$

The strain–life curve contains the dependency between a constant strain amplitude $\varepsilon_a$ and the achieved fatigue life in number of cycles until crack initiation N, Figure 3a. Both parts of this equation, the one describing the elastic part of the strain amplitude and the one describing the plastic part, appear as a straight line on a log-log scale. The lines intercept the values $\frac{\sigma'_f}{E}$ and $\varepsilon'_f$ at N = 0.5.

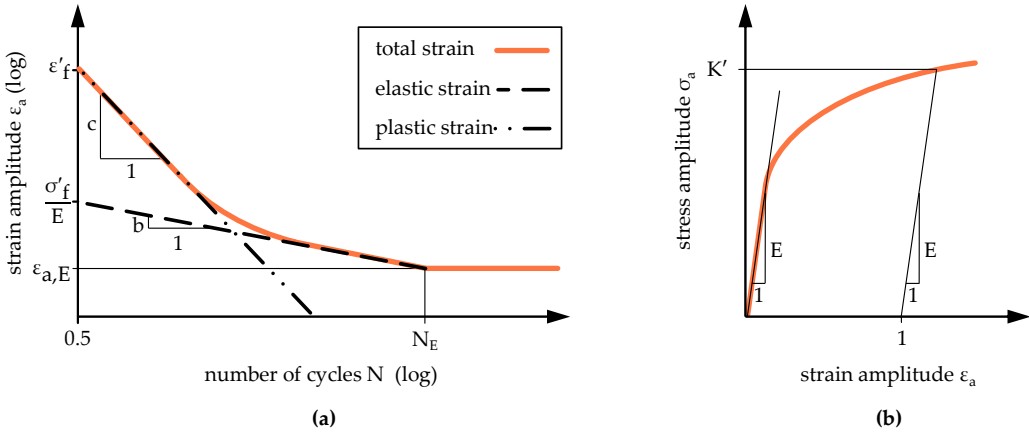

**Figure 3.** (**a**) Strain–life curve according to Coffin and Manson amended by fatigue strength $\varepsilon_{a,E}$ and (**b**) cyclic stress–strain curve according to Ramberg and Osgood.

The remaining cyclic properties K′ and n′ are used to determine the cyclic deformation behavior of a material using the approach of Ramberg and Osgood, [18], Figure 3b:

$$\varepsilon_a = \varepsilon_{a,el} + \varepsilon_{a,pl} = \frac{\sigma_a}{E} + \left(\frac{\sigma_a}{K'}\right)^{\frac{1}{n'}} \tag{2}$$

Since both Equations (1) and (2) contain the ratio of the elastic and plastic parts for one level of strain amplitude, the properties of both curves are not independent from each other. They are related by the following compatibility conditions:

$$n' = \frac{b}{c} \tag{3}$$

$$K' = \frac{\sigma'_f}{\left(\varepsilon'_f\right)^{n'}} \tag{4}$$

To determine the cyclic material properties, strain-controlled fatigue tests on unnotched specimens, Figure 2a, were conducted. To measure the strain, a clip-gauge extensometer with a measuring length of 5 mm was used. The strain amplitudes were chosen to be between 0.14% and 0.80%, so that the resulting numbers of load cycles covered the range of approximately 100 to $5 \cdot 10^6$. These tests were conducted in accordance with Standard SEP 1240 [19]. Since SEP 1240 is a German standard, it shall be described briefly: Strain-controlled fatigue tests are conducted at different levels. The strain amplitude is controlled and kept constant for the whole test. The axial reaction force acting on the specimen is measured and recorded throughout the test. The criterion used to identify the number of cycles for crack initiation N is a drop or increase in the measured stress amplitudes of 10% compared to the stabilized stress amplitude [19]. The result of a strain-controlled test is a data triplet of the strain amplitude, stress amplitude, and number of cycles endured. The determined number of cycles is usually valid for crack initiation. The stress amplitude $\sigma_a$, on the other hand, is determined at the half-life (N/2) of the specimen, where the material behavior has already stabilized after the so-called initial cyclic hardening or softening that occurs for many metallic materials. For a detailed explanation on the evaluation of such test results, see, e.g., [20]. The cyclic properties of the strain–life curve $\sigma'_f$, b, $\varepsilon'_f$, and c are determined by regression analysis of the individual test results (see SEP 1240 [19], also described in [20]). The properties of the cyclic stress–strain curves are calculated from the aforementioned properties using Equations (3) and (4).

In addition, stress-controlled staircase tests according to Standard DIN 50100 [21], also described in [22], were performed in the regime of the fatigue limit (very high cycle fatigue) on identical specimens as the strain-controlled tests. The determined fatigue limit can be added to the strain–life curve, Figure 3a. Therefore, the evaluated fatigue limit $\sigma_{a,E}$ needed to be converted from a stress to a strain $\varepsilon_{a,E}$, which was done by using the determined cyclic stress–strain curve, Equation (2).

To determine whether the deformation under cyclic loading localizes in certain areas of the layer structure, the local deformation behavior was investigated by drawing on a digital image correlation (DIC) system. Therefore, strain-controlled experiments with flat specimens (see Figure 2d) were carried out. The surfaces of specimens were prepared with a very fine black speckle pattern using white varnish as a primer coat to achieve high contrast. In particular, good adhesion between the specimen and the varnish as well as a thin application thickness were required for accurate measurements.

The double camera system ARAMIS 5M on an adjustable base together with software ARAMIS Professional 2017, both from GOM GmbH, Brunswick, Germany, were employed. Standard 50 mm lenses were sufficient to provide clear and detailed frames. Strain determination was performed within the ARAMIS software drawing on a distance between the evaluation points of 17 px and a facet size of 19 px. Thus, a facet comprises several pixels and characterizes a single evaluation point. This resulted in a geometrical distance of 0.208 mm between the evaluation points. The observation area on the specimen was thus composed of approximately 4000 evaluation points.

The tests were done at a strain amplitude of $\varepsilon_a$ = 0.4%, while a clip-gauge extensometer was used for strain control. Flat specimens were chosen to obtain clear readings with DIC. However, these types of specimen tend to buckle during compression. Therefore, an anti-buckling guide was needed for cyclic loading. While this was mounted on the specimen and covered its surface, no DIC recordings could be taken. After the deformation under cyclic loading stabilized, the test was stopped and the anti-buckling guide was removed. Then, the upper half of the hysteresis (no compression) was tested and the local strain was recorded with the DIC system, which enabled to get an impression of the local plastic strain amplitudes, Figure 4.

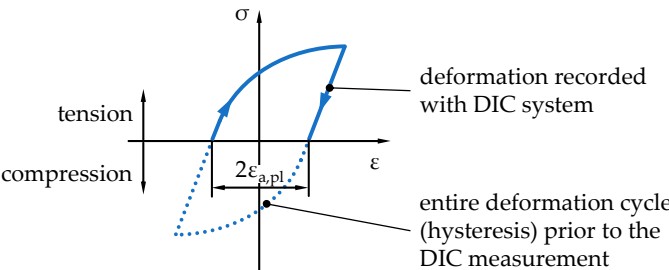

**Figure 4.** Locally forming hysteresis loops under cyclic deformation and measured strain in tension by digital image correlation (DIC; schematic).

## 3. Results

### 3.1. Microstructure and Local Distribution of Hardness

To gain a first impression of the influence of the interlayer temperature on the material properties, the microstructure was examined. The corresponding micrographs are shown in Figure 5. For both interlayer temperatures, it can be seen in region A that the appearance of the microstructure changes within the last three to five top layers between typical textures known from welds for the material group of steel and a rather fine grain structure. The finer grain size in the lower layers can be explained by the heat treatment through the sequel layers leading to a fine-grained microstructure due to the transition from ferrite to austenite and back in accordance to the formation mechanisms regarding heat-affected zones in joint welding. There, the temperature was kept at a higher level for a longer time than in the top layers for which sequel layers are missing.

For the lower layers, which were the focus of the following investigations, the microstructure is largely homogeneously distributed for both interlayer temperatures. Representative micrographs for this region are given in Figure 5 (see section C). On closer examination, it is found that the microstructure consists mostly of globular ferrite and pearlite for both interlayer temperatures. Due to the very long holding time at an elevated temperature, the first signs of dissolution may be seen in the pearlitic microstructural regions. Bainitic shares cannot be excluded and are not easy to distinguish from the pearlitic components. Optical determination of the phase fractions (digital threshold value correlation) showed an increase in the pearlite fraction by a factor of 1.6 from 15 to 24% when the interlayer temperature doubled. The increased share of perlite will lead to a larger distance between the $Fe_3C$ lamellas due to the very low carbon content of 0.07 mass-% in the material and is not expected to have a noticeable influence on the hardness. In order to verify this, the microhardness was measured over the centerlines of the examined cross-sections (see path B in Figure 5). The results for the upper parts of the cross-sections and the two interlayer temperatures are shown in Figure 5. In the non-normalized weld layers, the hardness values are slightly above 200 HV 0.2. In the normalized area, the hardness decreases to approximately 175 HV 0.2 on average. The latter value is also found in the lower (normalized) parts of the examined material (not shown here). The hardness seems to show largely homogeneously distributed values at about the same level for both interlayer temperatures. A distinguishable difference in hardness between layer and layer boundaries cannot be observed.

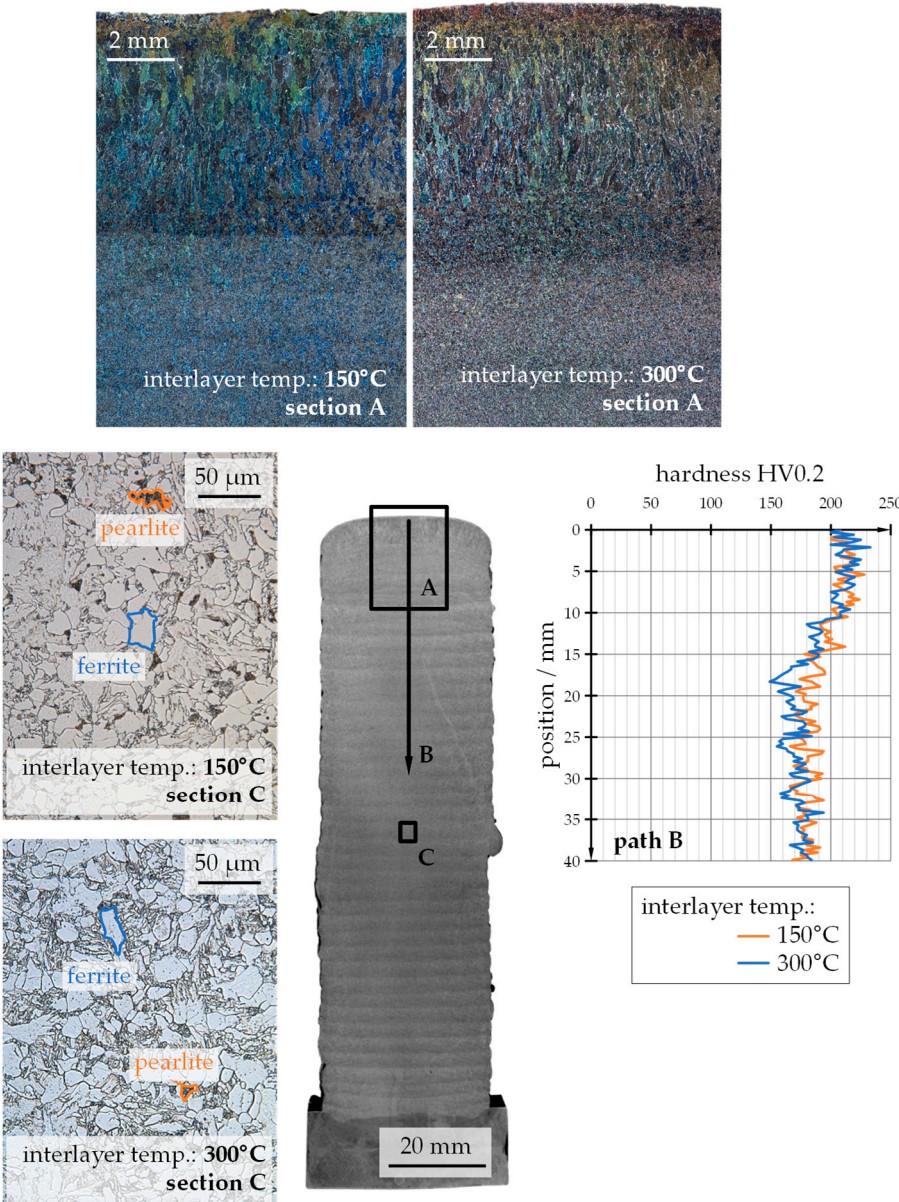

**Figure 5.** Micrographs and hardness measurements of cross-sections of WAAM material for interlayer temperatures of 150 and 300 °C.

## 3.2. Residual Stresses

To determine whether the interlayer temperature has an influence on the state of residual stresses within the manufactured WAAM material, residual stresses were measured on three cuboids of both interlayer temperatures at each of the three positions, top (D1), middle (D2), and bottom (D3), as shown in Figure 1c. Figure 6 shows the results of the measurement as maximum and minimum principal stresses for each measuring position and interlayer temperature. The results from the three cuboids of each interlayer temperature are grouped for each of the three measuring positions. In the lower part of the figure, the corresponding directions for the maximum principle stresses are given in relation to the later extraction directions crosswise and lengthwise.

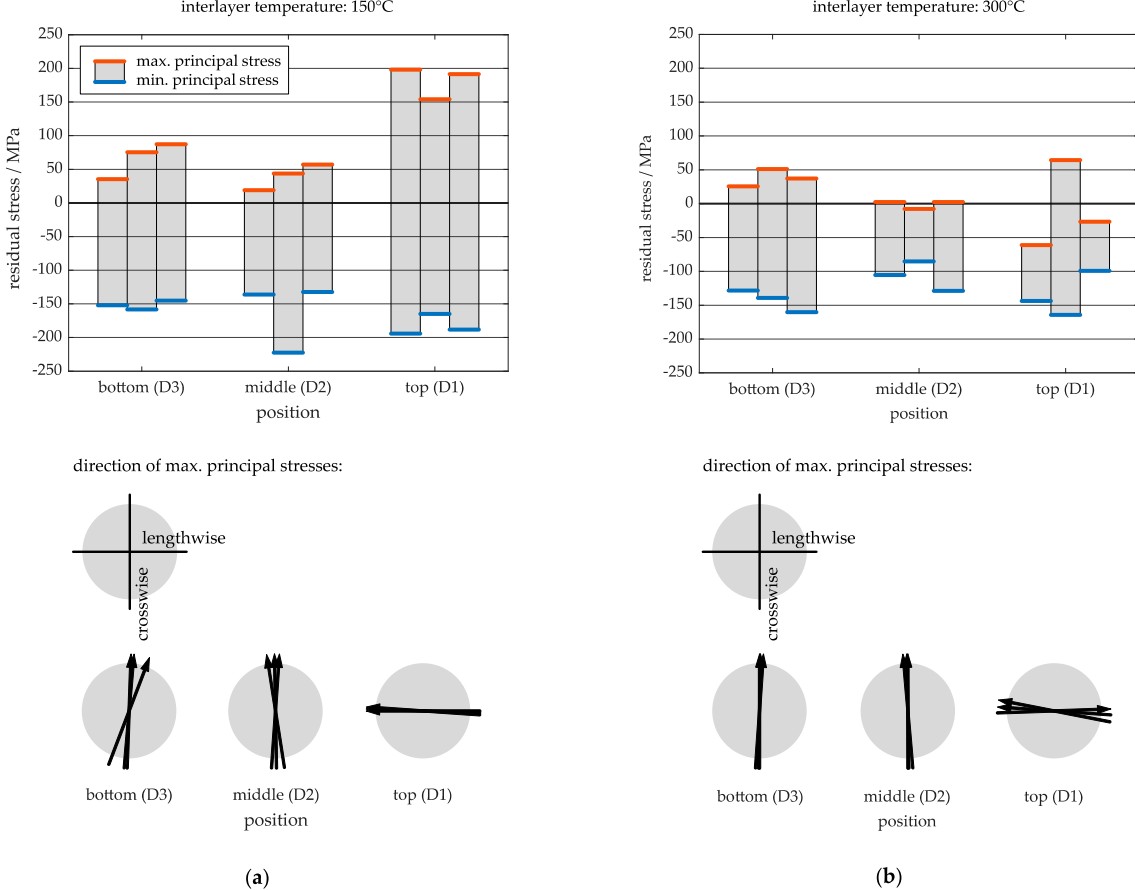

**Figure 6.** Residual stresses (principal stresses) and corresponding directions for the maximum principle stresses determined using the hole-drilling method for interlayer temperatures of 150 °C (**a**) and 300 °C (**b**).

It can be seen that

1.  tension and compression stresses are present within the material for both interlayer temperatures;
2.  for the normalized layers (measuring points D2 and D3) the residual stress states tend to be more on the compression side if the range between maximum and minimum principle stress is considered;
3.  the magnitude of the residual stresses and the spread between maximum and minimum principle stresses is, on average, lower for 300 °C than for 150 °C.

It can also be seen that the residual tension stresses are much higher at the top position (D1) of the cuboids manufactured at an interlayer temperature of 150 °C than in the lower regions of the same cuboids, as well as in the cuboids manufactured at 300 °C. The heat treatment through application of the top layers seems to reduce residual stress in the lower regions of the material. At 150 °C, the energy of the process seems to be insufficient to achieve this effect also within the top layers of the material compared to at 300 °C. In addition, the heat treatment through the layers later applied changes the direction of the principal stresses. The maximum principle stresses appear in a plane roughly parallel to the lengthwise extraction direction at the top position (D1), while at positions D2 and D3 the plane of the maximum principle stress is turned by about 90°.

### 3.3. Monotonic and Fatigue Properties

The aforementioned specimens used for the monotonic and fatigue tests were extracted below the five top layers that showed an influence through non-normalization (see Section 3.1). Therefore,

the strength properties discussed in the following sections are only valid for the normalized material mentioned above.

The monotonic tensile tests at room temperature as well as the fatigue tests were performed on the collected specimens lengthwise and crosswise to the manufacturing direction. Thus, the influence of the interlayer temperature, on the one hand, and the influence of the loading direction in relation to the welding direction, on the other hand, were demonstrated.

### 3.3.1. Monotonic Tensile Properties

The stress–strain curves of the monotonic tensile tests at room temperature for both the different interlayer temperatures as well as the extraction directions are shown in Figure 7. The corresponding average material properties determined are listed in Table 2. Table 2 also contains the standard deviations for these properties. However, it shall be pointed out that the given standard deviations are only based on four properties each, which limits the validity of this value.

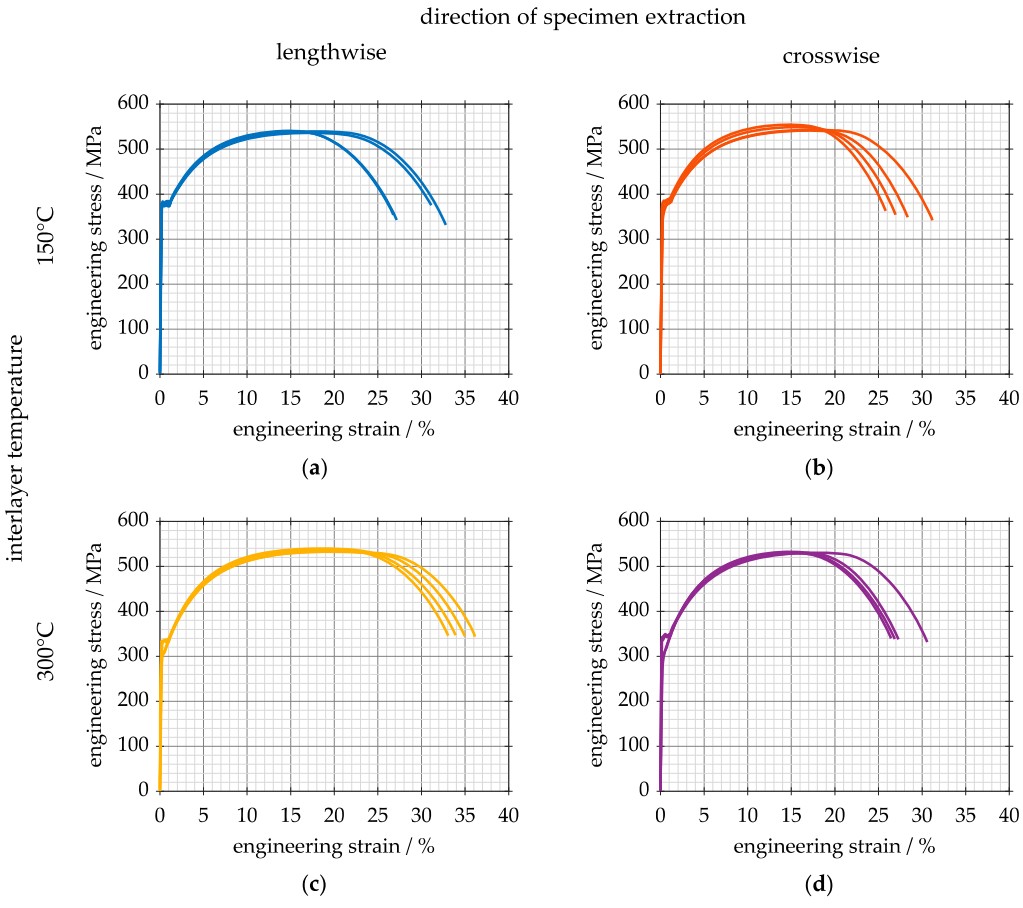

**Figure 7.** Stress–strain curves (engineering stresses and strains) from monotonic tensile tests at room temperature for interlayer temperature of 150 °C for lengthwise (**a**) and crosswise (**b**) extraction direction and for interlayer temperature of 300 °C for lengthwise (**c**) and crosswise (**d**) extraction direction.

**Table 2.** Average and standard deviation (values in brackets) of monotonic properties at room temperature.

| Interlayer Temp./°C | Extraction Direction | Tensile Strength $R_m$/MPa | | Upper Yield Strength $R_{eH}$/MPa | | Elongation at Rupture A | |
|---|---|---|---|---|---|---|---|
| 150 | lengthwise | 539 | (1.9) | 377 | (2.5) | 29.3% | (3.0%) |
| 150 | crosswise | 539 | (14.7) | 376 | (11.0) | 27.9% | (2.4%) |
| 300 | lengthwise | 536 | (3.3) | 319 | (16.7) | 34.3% | (1.3%) |
| 300 | crosswise | 531 | (1.2) | 325 | (19.2) | 27.6% | (1.9%) |

The tensile tests at room temperature contain two influences: the direction in which the samples are extracted and the interlayer temperature with which the material was manufactured. For this reason, the results will be compared below, one after the other, with regard to these two influences.

First, the direction in which the tensile specimens were extracted is considered in relation to the manufacturing direction. The monotonic strength, yield strength, and tensile strength do not show pronounced differences. The average elongation at rupture is noticeably larger for the interlayer temperature of 300 °C lengthwise to the manufacturing direction compared to in the crosswise direction.

By comparing the test results for the two different interlayer temperatures with each other, the following results are found: there is a significant difference in the strength properties only found for the yield strength, which can be related to less free interstitial carbon atoms present for a higher interlayer temperature, due to the larger amount of perlite. For the interlayer temperature of 150 °C, it is higher by about a factor of 1.15 than that at 300 °C. The tensile strength, however, does not show any discernible difference. It is about 536 MPa for all directions and interlayer temperatures.

In addition, tensile tests were conducted at elevated temperatures of 150 and 300 °C mainly to determine the yield strength at different temperatures for an assessment of the measured local residual stresses. Due to limited available WAAM material, tests at elevated temperatures were only conducted for specimens extracted in crosswise direction. These tests were done on another testing machine with a different clamping mechanism than the tests done at room temperature mentioned above. Therefore, a larger specimen diameter is used. To allow the results to be compared between the different testing conditions, tests at room temperature were also conducted at the testing machine for elevated temperatures. Since a clip-on extensometer was not available, the crosshead movement was used to calculate the necessary strains. Therefore, the elastic part of the stress–strain curves shown is spoiled for realistic elastic moduli, because the crosshead movement also contains the elastic deformation of the testing machine itself. However, the properties $R_{eH}$ or $R_{p0.2}$, $R_m$, and A could still be evaluated. For each case of test temperature and interlayer temperature, three tensile tests were conducted. The evaluated average tensile properties are given in Table 3 and the stress–strain curves are shown in Figure 8. Since only three tests were conducted for each average property, no standard deviation is calculated.

While a pronounced yield strength $R_{eH}$ is found at room temperature and 150 °C, at 300 °C the yield strength $R_{p0.2}$ is used instead (see italic values in Table 3). This need to be kept in mind for the comparison of the yield strength $R_p$—depending on the test temperature, the two different properties $R_{eH}$ and $R_{p0.2}$ are used. Figure 9 shows the dependencies between the determined average properties and the test temperature. To facilitate comparison, for room temperature, the properties determined from the stress–strain curves in Figure 7 have also been added to this diagram (triangle markers in Figure 9).

**Table 3.** Average monotonic properties at different temperatures in the crosswise extraction direction.

| Interlayer Temp./°C | Test Temp./°C | Tensile Strength $R_m$/MPa | Yield Strength $R_{eH}$/MPa or $R_{p0.2}$/MPa | Elongation at Rupture A |
|---|---|---|---|---|
| 150 | 20 | 549 | 385 | 31.3% |
| 150 | 150 | 503 | 364 | 23.6% |
| 150 | 300 | 585 | *358* | 24.1% |
| 300 | 20 | 542 | 343 | 29.2% |
| 300 | 150 | 477 | 319 | 26.7% |
| 300 | 300 | 550 | *318* | 23.9% |

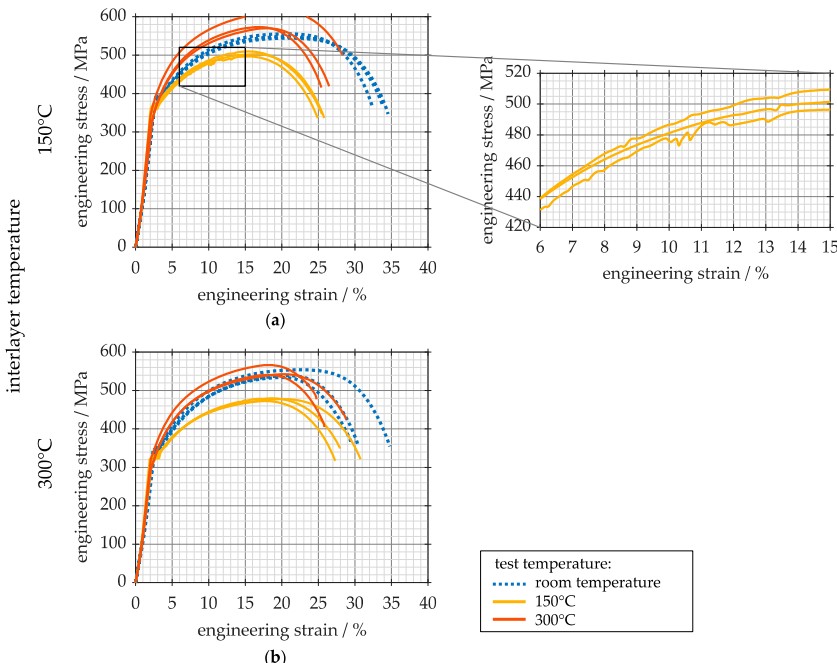

**Figure 8.** Stress–strain curves (engineering stresses and strains) from monotonic tensile tests for crosswise extraction direction and interlayer temperatures of 150 °C (**a**) and 300 °C (**b**).

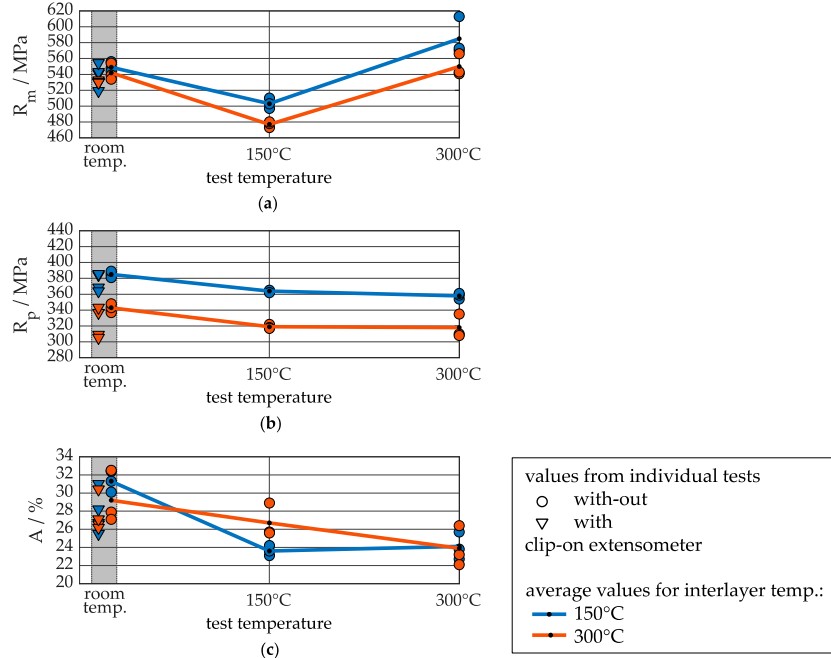

**Figure 9.** Average tensile properties for different test temperatures: (**a**) tensile strength, (**b**) yield strength, (**c**) elongation at rupture.

While for the yield strength, a slight decrease with increasing test temperature is found, the tensile strength shows lower values at 150 °C compared to at room temperature. However, an increase at 300 °C over the value at room temperature is observed. This tendency is found for both interlayer temperatures. It may be explained by effects of dynamic strain aging (DSA). DSA is a dynamical interaction between solute atoms and dislocations that may occur for certain strain rates and temperatures, [23]. It is known to appear for both ferritic and pearlitic structures and causes a local increase of tensile strength at about 350 °C. Above the temperature at which this maximum occurs, the tensile strength decreases with increasing temperature again, [24]. In addition, the Portevin-Le Chatelier (PLC; [25])

effect, often observed in association with DSA, could be witnessed at the test temperature of 150 °C. It manifests itself in kinks in the stress–strain curve, as shown in the enlargement of these curves in Figure 8a. The effect is observed at a test temperature of 150 °C for both interlayer temperatures, but is more pronounced for the interlayer temperature of 150 °C.

Due to the large scattering for the elongation at rupture, a clear dependency is hard to determine. However, it seems reasonable to state that there is a decreasing tendency with an increasing test temperature for both interlayer temperatures.

While no clear trend is found for the elongation at rupture between the two interlayer temperatures, the average strength properties $R_p$ and $R_m$ for the material manufactured with an interlayer temperature of 300 °C are below those for 150 °C for all testing temperatures. These differences are below a factor of 1.07 for $R_m$ and below a factor of 1.15 for $R_p$.

### 3.3.2. Fatigue Properties

This section describes the results from the strain-controlled fatigue tests in the low cycle fatigue regime as well as the stress-controlled staircase tests to determine the fatigue limit. Figure 10 shows the cyclic hardening/softening curves for three exemplary strain levels ($\varepsilon_a = 0.8\%$, 0.4%, and 0.2%) for each test series of strain-controlled tests. Moderate cyclic hardening is found independently from the strain level within the first load cycles applied. Afterwards, the stress amplitude shows an approximately stabilized behavior. For all specimens, a certain initial shift in the mean stress is observed. A mean compression stress of about −50 MPa is reduced towards zero as the number of cycles progress. The effect is slightly larger for the specimens that were extracted crosswise but seems to be independent from the interlayer temperature. This can be explained by the residual stresses mentioned above that remain within the microstructure to a small degree, despite the extraction of such small unnotched specimens.

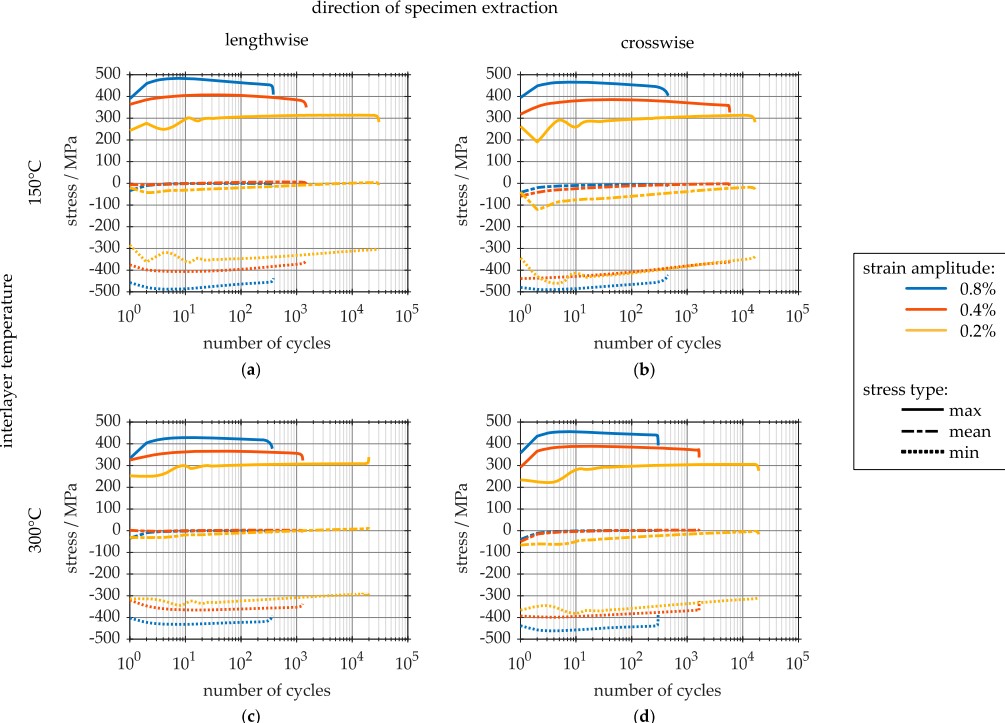

**Figure 10.** Cyclic hardening/softening curves for three exemplary strain amplitudes (semi-logarithmic scale) for interlayer temperature of 150 °C for lengthwise (**a**) and crosswise (**b**) extraction direction and for interlayer temperature of 300 °C for lengthwise (**c**) and crosswise (**d**) extraction direction.

The criterion used to identify the number of cycles for crack initiation N was a drop or increase in the measured stress amplitude of 10% compared to the stabilized stress amplitude, [19]. The appearance of a drop or increase in the stress amplitude depends on whether the crack formed inside or outside the measuring length of the clip-gauge.

To evaluate the cyclic properties, the elastic modulus needed to be specified. Here, a value of E = 206 GPa, which is in good agreement with both the monotonic and cyclic results, was chosen. The strain-life and cyclic stress–strain curves determined are shown in Figures 11 and 12 for all four test series. The corresponding cyclic properties are given in Table 4.

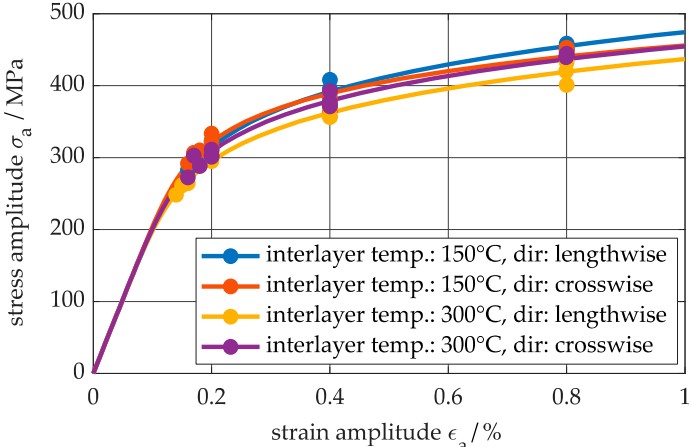

**Figure 11.** Cyclic stress–strain curves from strain-controlled tests.

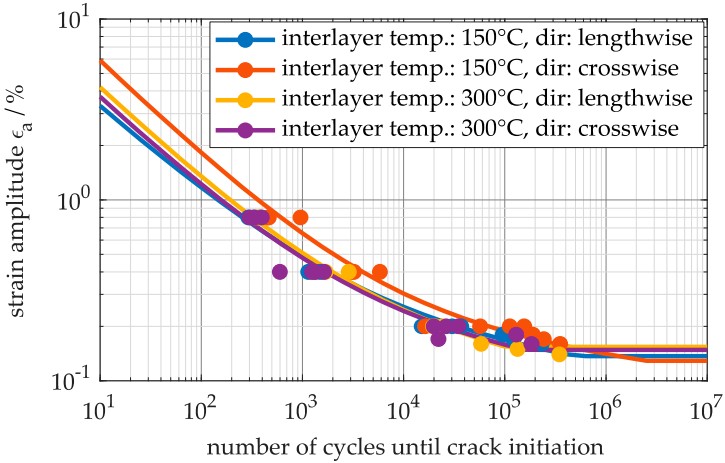

**Figure 12.** Strain–life curves (logarithmic scale) from strain-controlled tests including the fatigue limits from stress-controlled tests.

**Table 4.** Cyclic material properties.

| Interlayer Temp./°C | Extraction Direction | E/GPa | Strain–Life Curve | | | | Cyclic Stress–Strain Curve | |
|---|---|---|---|---|---|---|---|---|
| | | | $\sigma'_f$/MPa | b | $\varepsilon'_f$ | c | K′/MPa | n′ |
| 150 | lengthwise | 206 | 728.1 | −0.075 | 0.140 | −0.511 | 972.9 | 0.148 |
| 150 | crosswise | 206 | 707.4 | −0.067 | 0.292 | −0.550 | 821.3 | 0.121 |
| 300 | lengthwise | 206 | 707.7 | −0.081 | 0.199 | −0.542 | 900.6 | 0.149 |
| 300 | crosswise | 206 | 706.1 | −0.077 | 0.174 | −0.540 | 905.1 | 0.142 |

According to the definitions of the strain-life and cyclic stress–strain curves, Equations (1) and (2), the intercept points $\sigma'_f$, $\varepsilon'_f$, and $K'$ are located far from the center of the test results, Figure 3. $\sigma'_f$ and $\varepsilon'_f$ are defined at $N = 0.5$ and $K'$ at a plastic strain amplitude of $\varepsilon_{a,pl} = 100\%$. Therefore, these values are of limited suitability for a comparison between the different data sets. Small errors in the regression have a significant influence on the characteristic values themselves so that dependencies are hard to find. Therefore, the results are compared using the plots of the corresponding curves in Figures 11 and 12. It can be seen that the courses of the cyclic stress–strain curves are almost parallel to each other, and the curves are scattered over an area of approximately 50 MPa. Although the differences in the four curves are not very pronounced, the curves for the interlayer temperature of 300 °C are below the ones of the interlayer temperature of 150 °C. Apart from the curve of the interlayer temperature of 150 °C in the crosswise direction, all strain–life curves seem to fall in one scatter band. The first, however, shows numbers of cycles that are higher compared with the other curves by a factor of approximately 2.

The Coffin-Manson approach for the strain–life curve, Equation (1), originally does not contain a fatigue limit (endurance limit) and only applies to the low cycle and high cycle fatigue regime. In order to take the fatigue limit into account, stress-controlled tests were carried out using the staircase method with about 15 specimens each. The resulting fatigue limit values $\sigma_{a,E}$, Table 5, were converted from stress to strain by using cyclic stress–strain curves, Figure 11, and therefore may be included in the strain–life curves, Figure 12.

**Table 5.** Fatigue limits determined by staircase tests at $5 \times 10^6$ cycles, probability of failure $P_f = 50\%$.

| Interlayer Temp./°C | Extraction Direction | Fatigue Limit $\sigma_{a,E}$/MPa | Fatigue Limit Converted to Strain $\varepsilon_{a,E}$ |
|---|---|---|---|
| 150 | lengthwise | 256 | 0.137% |
| 150 | crosswise | 254 | 0.129% |
| 300 | lengthwise | 263 | 0.154% |
| 300 | crosswise | 267 | 0.148% |

It can be seen that the fatigue limit of the material manufactured with an interlayer temperature of 150 °C is slightly lower than that of material manufactured at 300 °C. However, the differences are within a factor of 1.05, at most.

Even though differences between the various interlayer temperatures and extraction directions are determined for both the cyclic deformation behavior and the fatigue strength, no pronounced trend is recognized from these. For example, the higher fatigue life of the crosswise extracted material with the interlayer temperature of 150 °C at the very end also results in the lowest fatigue strength of the four investigated groups of specimens. Thus, a preferred direction or interlayer temperature cannot be clearly identified.

### 3.4. Detailed Examination of Local Deformation Behaviour

Although the results shown previously suggest a relatively homogeneous distribution of the fairly anisotropic properties, the following must be critically noted: the diameters of the specimens, especially the one used for fatigue testing, roughly corresponded to the layer thickness. In the lengthwise direction, the specimens contain both the microstructure from the middle of a single layer and the boundary between two layers. Thus, several layers were tested in crosswise direction at once. During the strain-controlled tests, the strain was measured using a strain transducer with a measuring length of 5 mm. In cases where the microstructural constituents in the layers and layer boundaries showed different behaviors despite the results shown, it could be expected that strain localizations occurred and that the strain measured in the tests represented the average value over the measuring length. In order to rate this effect, the local deformation behavior was observed with a digital image correlation (DIC) system.

Therefore, strain-controlled experiments with flat specimens, Figure 2d, at a strain amplitude of $\varepsilon_a = 0.4\%$ were carried out. Figure 13a shows the used specimen in relation to the WAAM material. The layers are perpendicular to the loading direction. The recorded strain distributions are shown in Figure 13b for both interlayer temperatures.

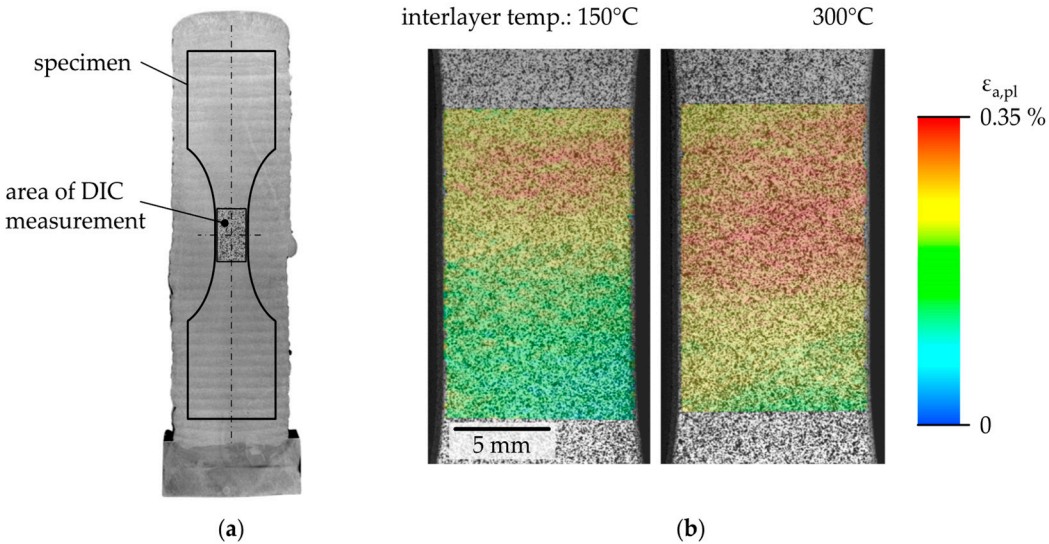

(**a**) (**b**)

**Figure 13.** Strain measurements with digital image correlation of flat specimens. (**a**) Specimen geometry and extraction position, (**b**) digital image correlation (DIC) results for the plastic strain amplitude within the parallel region of the specimen.

It can be noted that an inhomogeneous distribution of local plastic strain is observable for both interlayer temperatures. The size of the regions with different plastic strain amplitudes indicate that the individual layers differ in their deformation properties. However, a clear allocability to regions in the center of the layers and layer boundaries, as expected, is not found.

## 4. Discussion

### 4.1. Findings from the Conducted Experiments

First, the direction in which the tensile and fatigue specimens were extracted is considered in relation to the manufacturing direction. It can be stated that for the monotonic strength, yield strength, tensile strength, cyclic deformation, and fatigue behavior—all determined at room temperature—, no significant differences between the two directions can be detected. Only the average elongation at rupture for the interlayer temperature of 300 °C in the lengthwise direction and the fatigue life in the strain–life curve for the interlayer temperature of 150 °C in the crosswise direction stand out due to having higher values. The fact that there are no pronounced differences between the deformation and strength values in relation to the extraction direction indicates a largely homogeneous (although slightly anisotropic) microstructure despite the additive layer-by-layer structure of the investigated material. This is supported by the micrographs, which also show no difference in grain size or structure between the areas in the middle and edges of the single layers and therefore do not suggest pronounced directional dependency of the strength values.

Furthermore, the test results for the two different interlayer temperatures are compared with each other. The tensile strengths and the fatigue properties for the two interlayer temperatures show no distinguished differences. This is to be expected if the micrographic examinations and the hardness measurements are considered. The microstructure actually shows differences in the composition of ferrite and pearlite fractions for the two different interlayer temperatures. However, this is not expected in general to have an influence on the strength and the hardness also shows no appreciable dependencies. Differences in the monotonic strengths between the two interlayer temperatures are

only observed for the yield strength. The yield strength for the interlayer temperature of 150 °C is higher by a factor of about 1.15 than that at 300 °C. Even if no influence of the different structural compositions at the two interlayer temperatures could be found for the tensile strength, the different yield strengths can be explained very well by this: for the material manufactured with an interlayer temperature of 150 °C less, free interstitial carbon atoms are present, due to the larger amount of perlite. The higher amount of interstitial atoms leads to a higher pronounced yield strength for the material manufactured at 150 °C interlayer temperature.

The tensile tests at elevated temperatures reveal DSA and PLC effects that result in a local increase of the tensile strength at 300 °C test temperature and kinks in the stress–strain curves for 150 °C test temperature. The yield strength decreases with increasing test temperature, which is to be expected for metals. That this effect is clearly apparent for the tested temperatures that are identical to the used interlayer temperatures is worth mentioning for the following explanation of forming of residual stresses.

The residual stress states that occur at the two different interlayer temperatures shall be considered. First, the results of the three investigated measuring points and the different residual stress states in the cuboids are discussed, then the differences between the two interlayer temperatures are addressed.

In total, the layers with normalized material (measuring points D2 and D3) in the near-surface area show residual stress states that tend towards the compression side at both interlayer temperatures—the maximum principal stress is less far in the tensile range than the minimum principal stress in the compression range, Figure 6. This can be explained by the cooling conditions of the cuboid and is attributed to the cooling from the interlayer temperature level to room temperature. Since the cooling takes place from the surface inwards, the material is kept at a higher temperature level there for a longer time than at the surface. The yield strength, which is dependent from the temperature, is therefore lower in the core than at the cooled surface. This leads to a compression of the warmer core material compared to the colder surface layer during cooling and to the formation of residual compressive stresses when the entire body has cooled down [26].

Figure 6 shows the magnitude of the two principal stresses and their directions. The maximum principal stress (tensile residual stress) is present in a plane parallel to the layer structure, the minimum principal stress (compressive residual stress) under 90° to it, i.e., in crosswise direction. This also seems plausible. The compressive residual stresses are caused along the largest dimension of the cuboid by the cooling processes described above, and the tensile residual stresses by transverse contraction. This would also explain the ratios of the two principal stresses.

If, on the other hand, the residual stress states in the upper non-normalized layers (measuring point D1) are considered, different amounts of residual stress are noticed, but also a different orientation of the principal axis systems. These are rotated by 90° compared to those in the normalized layers. The residual stresses appear to be generated by cooling conditions of a different nature than those described above for larger cuboid areas. The cooling of material that has just been deposited and is now solidifying is initially carried out at a higher speed and locally more restricted. The internal stresses that are initially apparent in this way remain unchanged in the upper layers. In the lower layers, however, they are released by the subsequent layers applied later and the renewed energy input. The cooling from interlayer temperature to room temperature then follows the processes described above.

If the differences in the residual stress states for the two interlayer temperatures are now considered, it is noticeable that the principal stresses in the normalized layers (measuring points D2 and D3) are less far apart at 300 °C than at 150 °C interlayer temperature. The amounts of the two principal stresses are greater for 150 °C and are thus located further in both the tensile and compressive range. At measuring point D1 in the non-normalized range, significantly greater residual stresses can be detected for 150 °C than for 300 °C. The amounts of the latter are in the same range as the values in the normalized material. The higher interlayer temperature seems to significantly reduce the influence of the local cooling of the solidifying material due to longer holding at a higher temperature level.

The effects found here and the influence of the interlayer temperature can certainly be qualitatively transferred to a more complex WAAM geometry, i.e., a real component, but such a component will show a more complex thermal conductivity and cooling behavior due to the more complex geometry. The residual stress states in a real WAAM component will therefore be difficult to predict without simulation tools for this problem.

Overall, however, the positive effect of heat treatment by subsequent layers is noticeable, which tends to lead to compression-dominated residual stress states at the surface of a WAAM component. This is particularly desirable because fatigue failure in most cases originates from the surface and an already reducing influence on fatigue strength is to be expected in this area due to the very rough, unmachined surface. The fatigue strength-reducing effect of the rough surface would be at least partially compensated by the residual compressive stresses near the surface. At the same time, it should be noted that at low interlayer temperatures, considerable tensile residual stresses can be present in non-normalized layers.

Optical strain measurements over several layers showed that the distribution of local plastic strain amplitudes is inhomogeneous. However, an allocation of the strain localization to the areas of the individual layers and layer boundaries cannot be made. This can be interpreted in such a way that the material bonds well with the underlying material during application. The variation in local deformations could be due to fluctuations under cooling conditions, which are influenced by the thermal conditions, the feed rate, and the geometry of the already partially manufactured component. A more detailed examination of local inhomogeneities might be advantageous.

*4.2. Comparison of the WAAM Material with Conventional Materials*

One of the aims of this study is to compare the performance of the WAAM material under investigation to conventional steel material. First of all, the monotonic properties are examined. It could be expected that the generated material would reach properties within the requirements for weld materials, as specified in the standard for the welding filler material. However, except for the elongation at rupture, the WAAM material is not found to reach these specifications:

- The yield strength is smaller than the specification by a factor of 1.4 to 1.5;
- The tensile strength is slightly below the specified minimal value;
- The elongation at rupture is larger than the specified minimal value by a factor of about 1.6.

It can be stated that monotonic properties of WAAM material should not be expected to show the same properties as weld material from conventional weldments. This finding is not surprising as the manufacturing process and cooling conditions may vary considerably for the WAAM process.

Second, the performance of WAAM material concerning the fatigue life is compared. Since WAAM material is a new class of material and only few experimental findings are known for this class in general, a rating of the fatigue performance can only be done by looking at conventional steel materials. For this, a study by Wächter and Esderts [27] that determined the average statistical dependencies between the monotonic tensile strength and cyclic properties on a large database of experimental results is considered. From that study, a method for estimating cyclic properties originated, amongst others, for steel materials (Table 6).

The estimation method just needs the tensile strength $R_m$ to estimate the cyclic properties of the strain–life curve. For steel, two subgroups are distinguished: steel (rolled and forged) and cast steel. To get an insight of how the WAAM material under investigation behaves in relation to other steel materials, the cyclic material characteristics are estimated from the tensile strength of the WAAM material using the method shown in Table 6. The determined average values for the $R_m$ of the WAAM materials vary over a range from 531 to 539 MPa, Table 2. This range has no noticeable influence on the estimated values when plotted in a diagram. Therefore, the average tensile strength $R_m$ = 536 MPa is used for the estimation. The estimated strain–life curves are plotted in Figure 14 beside the results of the strain-controlled tests.

**Table 6.** Method for estimating cyclic properties (for the material groups steel and cast steel) [27].

| Property | Steel | Cast Steel |
|---|---|---|
| $\sigma'_f$ | $3.1148 \text{ MPa} \cdot \left(\frac{R_m}{\text{MPa}}\right)^{0.897}$ | $1.732 \text{ MPa} \cdot \left(\frac{R_m}{\text{MPa}}\right)^{0.982}$ |
| b | −0.097 | −0.102 |
| E | 206 GPa | 206 GPa |
| $\varepsilon'_f$ | $\min\left(\begin{array}{c} 0.338 \\ 1033 \cdot \left(\frac{R_m}{\text{MPa}}\right)^{-1.235} \end{array}\right)$ | $0847 \cdot \left(\frac{R_m}{\text{MPa}}\right)^{-0.181}$ |
| c | −0.52 | −0.58 |
| Range of validity | $R_m = 121 \ldots 2296$ MPa | $R_m = 496 \ldots 1144$ MPa |

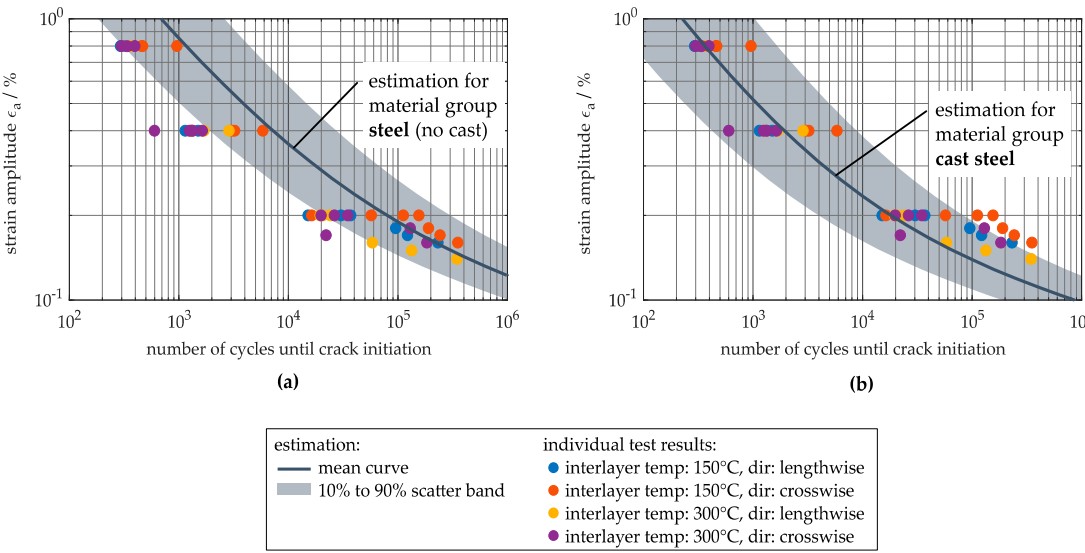

**Figure 14.** Estimated strain–life curves for steel (**a**) and cast steel (**b**) for $R_m = 538$ MPa and individual test results from Figure 12.

The study presented in [27] also contains a deviation range between the 10% and 90% quantiles for the estimation method. This scatter parameter describes the scattering of experimental values compared with the estimated curves and is also included in Figure 14. It can be seen that the fatigue life of the WAAM material is below the estimated strain–life curve for rolled and forged steel material in most experimental results, while it fits the estimated curve and rated scatter band for cast steel quite well.

With both the estimated values for the strain–life curve as well as Equations (3) and (4), it is possible to estimate the properties for the cyclic stress–strain curves. These are plotted in the same way as the strain–life curve, Figure 15. Since the two estimated curves do not differ much between steel and cast steel, the deformation behavior for the WAAM material is in good agreement with both material groups.

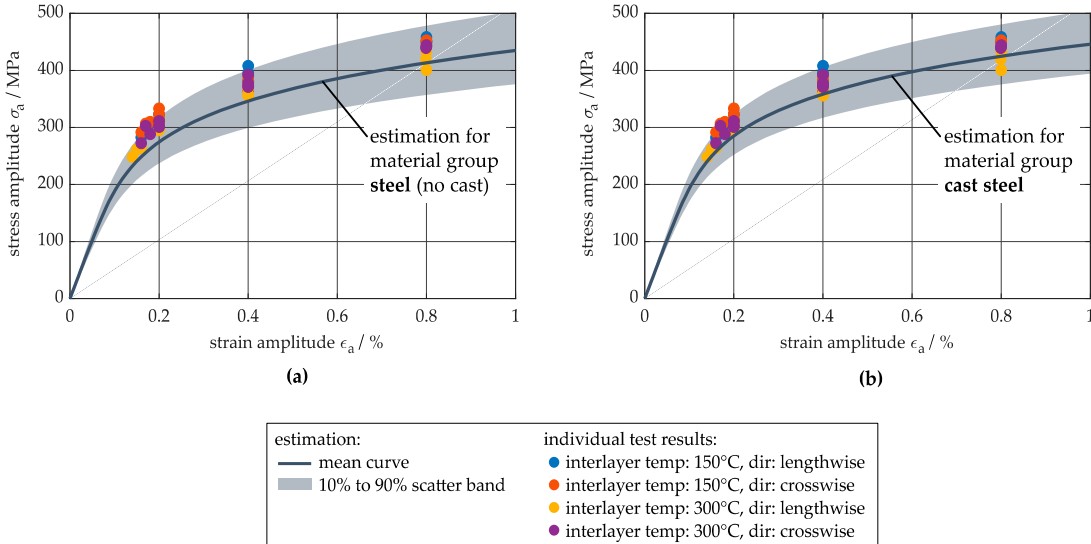

estimation:
— mean curve
▓ 10% to 90% scatter band

individual test results:
● interlayer temp: 150°C, dir: lengthwise
● interlayer temp: 150°C, dir: crosswise
● interlayer temp: 300°C, dir: lengthwise
● interlayer temp: 300°C, dir: crosswise

**Figure 15.** Estimated cyclic stress–strain curves for steel (**a**) and cast steel (**b**) for $R_m$ = 538 MPa and individual test results from Figure 11.

## 5. Conclusions and Outlook

Monotonic and cyclic properties for steel material with an average tensile strength of $R_m$ = 536 MPa were determined in this study. These properties show a slight anisotropy but do not seem to be noticeably distributed between layer and layer boundaries. The monotonic properties of the investigated WAAM material differ significantly from the values expected from the same weld filler material in conventional welding use. The measured cyclic properties related to the tensile strength are in better agreement with cast steel than with rolled and forged steel.

Optical strain measurements over several layers show that the plastic strain amplitudes are not distributed homogeneously. The recognizable areas with homogeneous behavior are larger than one single layer. This influences the local stresses and strains that occur in a WAAM component, which makes them challenging to predict. However, knowledge of the local stress and strain amplitude is one of the most important influences on a fatigue life calculation. This aspect needs to be examined more closely in the future.

The test results for the investigated material and manufacturing process show that it is possible to increase the interlayer temperature from 150 to 300 °C without decisively reducing the performance in terms of monotonic and cyclic strength. The yield strength decreases slightly when the interlayer temperature is increased. Overall, a higher interlayer temperature can lead to a considerable reduction in manufacturing time (e.g., factor 2 for the cuboids manufactured for this study), more flexible planning of the process, and, thereby, improved economic efficiency of the WAAM process.

It must be noted that the results presented here are only a first step towards the characterization of medium strength WAAM materials. The manufacturing process itself must be pushed to the limits of what is feasible in order to be as efficient as possible. For example, the question must be answered of at which interlayer temperatures—even higher than the ones investigated here—a component cannot be manufactured anymore, since the melt no longer solidifies with sufficient dimensional stability. In addition to the strength of the base material, other influencing variables such as surface texture and mean-stress sensitivity must be taken into account in order to be able to carry out a qualified fatigue assessment of WAAM components.

**Supplementary Materials:** The following are available online at http://www.mdpi.com/2076-3417/10/15/5238/s1, Table S1. Individual test results of strain-controlled tests, Figure S1. Test schemes for stress-controlled staircase tests according to DIN 50100, Drawing 1. Technical drawing of specimen for fatigue tests, measurements in mm, Drawing 2. Technical drawing of specimen for monotonic tensile tests at room temperature, measurements in mm, Drawing 3. Technical drawing of specimen for monotonic tensile tests at elevated temperatures, measurements

in mm, Drawing 4. Technical drawing of specimen for strain measurements with digital image correlation, measurements in mm.

**Author Contributions:** Conceptualization, M.W. and K.T.; manufacturing of raw WAAM material, K.T., S.K., and V.W.; monotonic tensile tests at room temperature and cyclic tests, M.H., L.M., and A.E.; monotonic test at elevated temperatures, M.L. and V.W.; local strain measurements with DIC, C.L., S.H., and L.M.; writing—original draft preparation, M.W.; writing—review and editing, M.W., C.L., S.H., L.M., M.L., M.H., and K.T.; supervision, A.E., V.W., and S.H. All authors have read and agreed to the published version of the manuscript.

**Funding:** This research received no external funding. However, we acknowledge support by the Open Access Publishing Fund of Clausthal University of Technology.

**Conflicts of Interest:** The authors declare no conflicts of interest.

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
