# Peer review of "Monotonic and Fatigue Properties of Steel Material Manufactured by Wire Arc Additive Manufacturing"

_applsci, doi:10.3390/app10155238_

Round 1
Reviewer 1 Report
Line 105 – What was used as the acquisition system?
Line 109 - a graph with the temperature’s vs time would be useful
Fig 5 the scale should start at 100 vickers to be more perceptive
Line 173 – Which was the values of strain amplitude chosen?
Line 179 - Typo - “amplitudeamplitudes”
Line 222 – What do you mean with the “quality of the microstructure”. The microstructural properties are clearly different, but its quality only depends of a possible application.
Line 237 – How do you determine the perlite fraction?
Line 247 - The chapter 3.2 Residual stresses are mostly presented as a results report, the it should be more discussed. Even in the discussion chapter the residual stress results are presented and not discussed
The author say in line 343 that no differences in the monotonic properties were found between the interlayer temperature, but there are nothing saying why that happen.
The reason to perform uniaxial tensile tests at 150 ͦC and 300 ͦC are not presented, and the results of these tests are inconclusive and not discussed.
Line 385 “the curves for the interlayer temperature of 300 °C are below the ones of the interlayer temperature of 150 °C” What does that mean? Why does that happen? You must discuss the results from a scientific point, otherwise this is a materials properties report and not a scientific research.
Line 439 “crosswise direction stand out due to having better values” What are better values?
Line 443 “This is supported by the shown micrographs, which suggest that the microstructure found in WAAM differs significantly from the ones known from conventional welding processes.” The differences between the microstructure of WAAM and Welding are well know on the literature.
447 “The expected higher monotonic strengths at lower interlayer temperatures are only observed for the yield strength” - Why was that expected?
450 – “Since the tensile strength of a material often shows correlations to fatigue strength parameters, it is less surprising than originally expected that these parameters do not differ significantly between the two interlayer temperatures investigated. Furthermore, it is not surprising that a higher interlayer temperature is found to have a positive effect on the residual stress state.”
I understand that you have been surprised by that, but this is not a results discussion…
464 – “The variation in local deformations could be due to fluctuations under cooling conditions, which are influenced by the environmental conditions, the feed rate, and the geometry of the already partially manufactured component. A more detailed examination of local inhomogeneities might be advantageous.”
I do not believe that the “environmental conditions” has significative influence in the cooling process.
Reviewer 2 Report
The article presented monotonic and cyclic properties of stainless materials manufactured by wire arc additive manufacturing. The project is well done and paper is well written.
Author Response
Dear Reviewer,
Thank you very much for the review of our manuscript. We very much appreciate the good rating of our paper.
Best regards!

Reviewer 3 Report
The study describes monotonic and fatigue properties of steel material manufactured by Wire Arc Additive Manufacturing.
These are important practical studies. In my opinion, all stages of research and analysis are well described. The conclusions are formulated in a concise and informative manner.
Author Response

(The authors gave the same response as above.)

Reviewer 4 Report
General comment:
The reviewed article concerns the issue of assessing the strength properties of a material obtained in the Wire Arc Additive Manufacturing process. The content of the article refers to very current issues of effective use of constructional material while minimizing material losses. An important problem is the assessment of the properties of the material thus obtained and the impact of manufacturing technology on the change in properties in relation to the base material.
The paper presents the process of forming material using WAAM. The procedures of the experiments and their results have been described in detail and discussed. The authors used standard static and fatigue assessment procedures. The summary refers the properties of the material obtained in WAAM technology with the base material.
In my opinion, the article is prepared very thoroughly in terms of content in relation to experimental research. The paper presents important issues for current research problems, but does not contain significant scientific aspects.
Comment to text editing:
Referring to the content of the article, some editorial inaccuracies were noted:
line 84 - "we describe", I recommend using the form "... Base on this, secondly, the used speimens .... raw material are presented.
line 90 - "... in accordance with [10]". Reference number should not be part of a sentence. I suggest "... with ISO Standard 14341 [10]. Similarly, the text in lines 142, 144, 147, 148, 175. A similar problem occurred in lines 184-186. Please, rebuild sentences.
line 188 - I suggest "In addition, stress-controlled staircase tests [21, 22] were performed ...
lines 331, 332, 361 - remove "see e.g." or "see"
lines 91, 92 - please enter the symbols as they are used in the literature of the subject of fatigue of materials, e.g. Rm ->σU - ultimate strength, Rp,min->σy,min - minimum yield strength
line 301 - Table 2 - insert coma "," instead of "in". Please enter parameters σU, σy. ReH ->?
Please, enter uniform symbols for parameters for the entire text: Table 2, line 311, 315. If ReH = Rp0.2 = σy, too many symbols create ambiguity.
lines 207-209 : Figure 4, meanwhile, the first reference to Figure 4 is in line 217. The figure should appear after the first reference in the text.
line 248 Chapter 3.2 Residual stresses begins with Figure 6. Please move the drawing after the paragraph containing the reference to Figure 6.
lines 293-297: I think this paragraph desribes figure 8, please, introduce reference to figure 8, e.g. at the end of first sentence in paragraph (line 295).
line 345 - Chapter 3.3.2. begins with figure 10. Please, move the figure 10 after first paragraph (lines349-359)
line 346 - figure 10, Please, consider to remove annotation "log" in axis description. In the description under the figure, it can be added in brackets (semilogarythmic coordinates).
lines 369, 371 - figures 11 and 12, insert coma "," instead of "in". Please, consider to remove "log" in axes description (fig. 12), add description (bilogarithmic coordinates) in figure caption.
line 393 - σaf is the symbol for fatigue limit, similar εaf. Please consider making a change, including Table 5 and Figure 3a.
line 469 - Please, remove ", as stated in Section 1,"
Final remark:
The comments indicated do not reduce the value of the work, but in my opinion they will make the text more coherent.
I highly appreciate the cognitive value of the article and I propose to publish the article after minor revision.
Reviewer 5 Report
Dear authors,
thank you for this interesting article. I'd like to mention some trifle, taht should be considered to change:
line 16 The latter... it is not clear to what this "latter" refers
line 23 ...were also considered. Not the best choice of words, since hardness and residual stresses were measured and compared.
line 24 ...material, we aimed... change into "it was aimed"?
line 44 prototypes instead of individual pieces?
line 51-55 the two aspects both mix up properties and economics, why don't you formulate one economic aspect and one concerning the properties
line 57 ...to a very limited extent or individual cases... in this paper there has also only been one material investigated. It would be much more interesting to read which materials have been investigated by different authors and what results they received (same properties, increase in strength, decrease in fatigue strength...)
line 90 add the material teh welding material is usually applied for
line 143 why HV 0.2? Measurements with higher loads would cover up bigger microstructural areas. Could be of advantage.
line 153ff Calculation of strain from the movement of the crosshead bears many failures. Since there is a pronounced yield strength the strain is not important for the strength values. The elongation after rupture should be measured at the ruptured sample in this case.
line 179 amplitudes
line 216 ..,which enables us --> which enables to get an impression...
line 225 I am quite sure that the heat treatment through the sequel layers does not lead to recrystallisation, since there is no deformation taking place in the process, which is a indispensable element in recrystallisation. Due to the welding process you'll certainly exceed the austenitisation temperature and therewith normalize the microstructure (multiple). This causes a significant grain refinement - Please correct this error also in the rest of the document. Afterwards the microstructure is softannealed bs the following temperatures beneath austenitization temperature. This leads to resolution of pearlite lamellae.
line 238 is the difference in the pearlite fraction resilient and/or relevant if the hardnesss of the microstructur does not differ?
Figure 6 represantation of principal stresses is not well chosen. The spread (min-max) should end at the min and the max value respectively. With max values below zero the disposition otherwise gives a wrong picture
line 264 4. the spread of the residual stresses is smaller for a higher interlayer temperature
line 273 ---> keep in mind for discussion: What does this mean for components directly produced with WAAM (not milled out of a cuboid)
line 291 Delete the sentence in brackets but calculate mean value and standard deviation for the measured results! At least there can already be seen an significant tendency for the change in interlayer temperature.
---> keep in mind for discussion: Discuss the influence of free interstitional atoms on the yield strength.
Table 2: mean value and standard deviation, please!
Figure 8/table 3: in line 118 is noted that samples were taken in lenth- and crosswise direction. Where are the results of the lengthwise taken samples
Figure 9: error bars missing
line 464: check sentence: This might be interpreted...
sounds weird
line432 Discussion: Apart from the points already mentioned above, I'd be interested in the discussion how an even higher interlayer temperatur might affect the results. Furthermore please remember that in production you'd normally wouldn't produce a cuboid and mill samples out of it. How will this affect the properties? What if the structures are partly more filigree?
References:
13. Strain-Gauge Method
15. Did you choose Satndard EN ISO 6892-1:2016 on purpose? Theresanewersion from 2019. If you fulfil the requirements you should refer to the actual version
I hope I could help to improve your paper.
Kind regards,
Round 2
Reviewer 1 Report
The author adressed all de major comments and i think that the paper is now ready for publication.